# Fully Ossified Thyroid Cartilage Found among the Skeletal Remains of A 21-Year-Old Slavic Soldier: Interpretation of a Case

**Alessia Leggio *** [ID] **and Francesco Introna**

Department of Interdisciplinary Medicine, Section of Legal Medicine, Policlinico di Bari Hospital, University of Bari, Piazza Giulio Cesare 11, 70124 Bari, Italy; francesco.introna@uniba.it
* Correspondence: alessialeggio11@gmail.com

**Abstract:** The degree of ossification of the thyroid cartilage in anthropological studies is related to other methods of determining the age of a skeleton. The endochondral ossification process begins at the age of approximately 15–20 years and is generally completed in the fifth or sixth decade of life. In the present case, early and complete mineralization of the thyroid cartilage of a skeleton belonging to a 21-year-old young soldier who died in 1946 was observed. Ossified thyroid cartilage at a very young age is rare and may be associated with specific symptoms, such as compression due to trauma, or may also be related to hormonal dysfunction and various diseases that may cause early mineralization. A macroscopic morphological evaluation and radiographic examination of the thyroid cartilage were performed and a decalcification test was applied to a sample taken from the thyroid cartilage to confirm that the structure was indeed mineralized. There is nothing to exclude that this is simply a physiological anatomical variation. Knowledge of this rare anatomical abnormality at a young age would be useful for the diagnosis of various pathological conditions.

**Keywords:** anthropology; skeletal remains; thyroid cartilage; determination of age at death; Slavic soldiers

## 1. Introduction

The thyroid cartilage is shield-shaped, located in the anterosuperior part of the larynx and lateral to it, and is one of the largest laryngeal cartilages. Consisting of two quadrangular laminae that join together anteriorly at an open angle on the dorsum, this structure re-flexes its function to protect the delicate structures of the glottis [1,2], particularly the vocal cords. The skeleton of the thyroid cartilage consists essentially of hyaline cartilage [3], which is not vascularized.

The penetration of the vessels from the perichondrium to the thyroid cartilage is the prelude to the endochondral ossification process, which begins around 15–20 years of age and is generally completed in the fifth or sixth decade of life, and this mineralization process occurs in males as opposed to women [4].

Several studies from 1974 and 1980 identified a correlation between the degree of ossification of this structure and age [5], describing different stages of mineralization over time [6].

In 1983, (Figure 1) on the basis of these studies, the ossification process of the thyroid cartilage was divided into nine stages, distinguished by age groups [7,8]:

Often, during anthropological investigations for age determination, the degrees of fusion of the ossification centers of the skeleton are analyzed, and to date, this includes that of the thyroid cartilage. In our case series, applying this method, a degree of total ossification was found on a skeletal remnant belonging to a 21-year-old young soldier.

Complete ossification of thyroid cartilage at a very young age is rare and may be related to specific pathological conditions. Among the many hypotheses that cause premature ossification at a young age, two stand out. Firstly, ossification due to trauma or

contusion of the cartilage structure. Secondly, it could be linked to hormonal dysfunction, causing premature mineralization of the structure.

- **Stage 1**: 15-17 years, the first ossification center appears in the lower portion.
- **Stage 2**: 18-21 years, the ossification centers at the base of the inferior horns fuse with those near the inferior thyroid tubercle forming the so-called posteroinferior triangle.
- **Stage 3**: 21-26 years, complete ossification of the lower horns.
- **Stage 4**: 31-35 years, the upper horns ossify.
- **Stage 5**: 28-39 years, the ossification centers of the left and right lower portions of the cartilage body weld together.
- **Stage 6**: 38-45 years, the paramedian process is completed.
- **Stage 7**: 48-53 years, the median process and a posterior opening in the cartilage body are formed.
- **Stage 8**: 51-58 years, the fusion of the superior thyroid tubercles and median process.
- **Stage 9**: 57-68 years, the ossification of the right and left lamina is completed and a second opening in the body of the thyroid cartilage is formed.

**Figure 1.** Classification ossification thyroid cartilage Černý 1983.

Despite the two valid hypotheses, today, the causes of this anomaly remain unknown, and knowledge of this anatomical anomaly, which is rare at a young age, would be useful for the diagnosis of various pathological conditions.

## 2. Case Report

In 2019, the Institute of Legal Medicine of Bari began a research project at the Ossuary of the Monumental Cemetery of Bari that led to the discovery of 93 skeletal remains of Yugoslavian origin. Looking at the available historical documents and linking the evidence of objects inside the boxes containing the finds, it was possible to confirm that the skeletal remains belonged to soldiers of the Royal Yugoslav Army—of the national movement [9] of Ravna Gora—in exile in 1941.

The Chetniks were led by Serbian General Dragoljub "Draža" Mihailović, an anti-communist, anti-Nazi, nationalist, and monarchist force, in opposition to the partisan resistance led by the communists of the famous commander Josip Broz Tito at the outbreak of World War II. During the anthropological analysis of skeletal remains, the skeleton of a young 21-year-old Slavic soldier was examined. The recovered skeleton is 80% intact with some incomplete portions [10], and fully ossified thyroid cartilage was detected.

In order to determine the biological profile of the young Slavic soldier, a complete anthropological analysis of the skeleton was carried out to determine the sex, age, and stature. Subsequently, in order to ascertain the anomaly related to the early mineralization of the cartilage, a macroscopic morphological evaluation, a radiographic examination, and a decalcification test performed on a sample taken from the structure were carried out to confirm the actual mineralization. Morphological criteria were used for sex determination for both the skull, complete with mandible and hip [11], confirming a degree of male sexualization. Age at death was determined by applying the method of the degree of epiphyseal fusion of the entire skeletal system and the degree of fusion of the cranial sutures [12,13]. The method of the degree of dental wear by dental analysis and the study of the morphology of the pubic symphysis completed the estimation of the age at death [14,15]. For the determination of stature, the methods of Trotter and Gleser were used [16].

Together, the various methods used determined a male subject, with age at death between 20 and 25 years and a height of 1.69 ± 3.5. Due to the rare abnormality, the thyroid cartilage was subjected to several investigations (Figure 2).

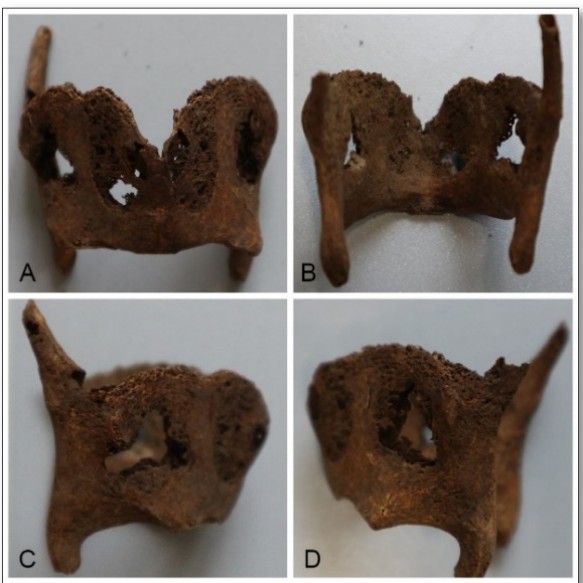

**Figure 2.** Thyroid cartilage projections. (**A**,**B**) Frontal projections: (**A**) Anterior, (**B**) posterior; (**C**,**D**) lateral projections: (**C**) Right, (**D**) left.

Firstly, morphological examination, with macroscopic observation [6,7] to determine the age of the subject, which, in the present case, was classified as stage 9 (57–68 years) given its complete mineralization (Figure 3).

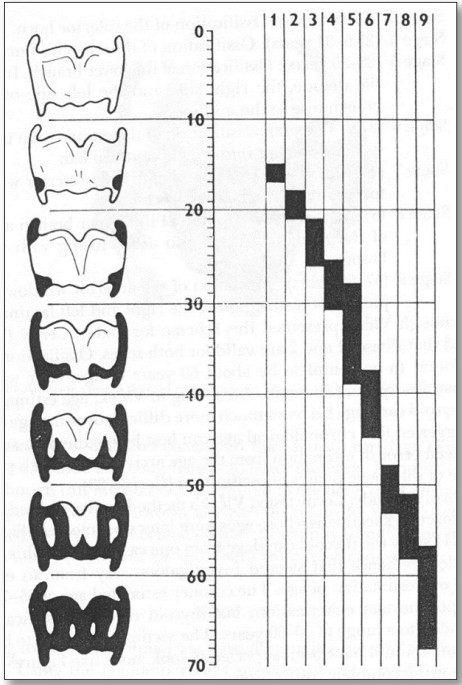

**Figure 3.** Ossification degree of the thyroid Cartilage by Černý 1983.

The second investigation was radiological to ascertain the complete mineralization of the right and left laminae by detecting the degree of ossification (Figure 4). The results confirmed the partial degree of ossification, which left out the "internal windows", typical of the attributed male morphology. The third investigation, that of the sample decalcification test, confirmed the results of the radiological examination.

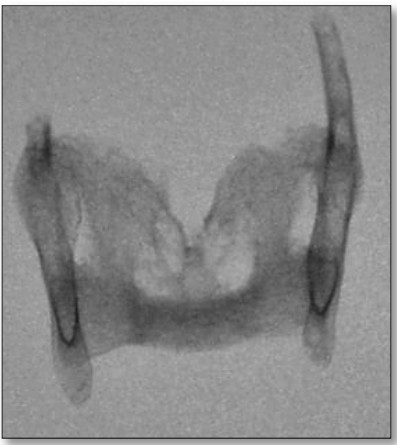

**Figure 4.** Thyroid cartilage Radiography—Frontal Projection.

A sample of the thyroid cartilage under examination was subjected to a decalcification test [17] using 20% hydrochloric acid for a period of 3–5 days. Hydrochloric acid acts with mineral salts. In this specific case, it acts with calcium carbonate and calcium phosphate, which are the main mineral components of bone. By acting with such minerals, the test confirms complete ossification; otherwise, no reaction would be observed (Figure 5).

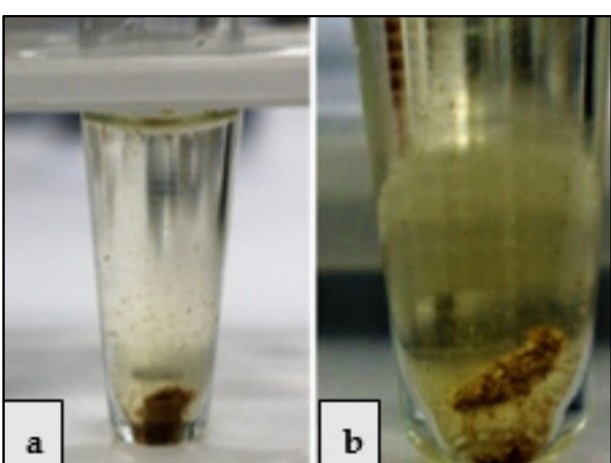

**Figure 5.** Decalcification of a thyroid cartilage fragment in 20% HCl. (**a**), descaling on the first day. (**b**), descaling on the third day.

## 3. Discussion

Due to aging, thyroid cartilage tends to calcify [18]. With the passage of time, it is converted into bone tissue through the process of endochondral ossification until complete ossification in 57–68 year olds.

The degree of ossification is lower in females than in males, but there is symmetry in the ossification of the thyroid cartilage in both sexes. This phenomenon is rarely observed, and for these reasons, the total mechanism of mineralization of this cartilage is not well understood [19]. During the mineralization process, the hyaline thyroid cartilage shows a progressive increase in collagen fibers, which over time, calcify until they become completely ossified [20]. The mineralization process starts near the posterior-inferior border of the lamina and then extends into the inferior horn and then towards the superior-prior horn. Finally, it spreads completely [7]. Many studies state and demonstrate that the degree of ossification is divided into nine stages, and each stage corresponds to a specific age [8]. In the present case, a general anthropological survey of the skeletal remains was

carried out to analyze the biological profile, determining a male subject with age at death of 20–25 years and stature of $1.69 \pm 3.5$.

Regarding the ossification of the thyroid cartilage, a morphological investigation was carried out with macroscopic observation of the ossified structure, noting that ossification is almost complete except in two places. Two small openings, so-called "internal windows" are visible in the posterosuperior and antero-inferior regions of each lamina. In most cases, commonly, there may be one or two windows of an unoxidized or less ossified zone in the center of each lamina. In fact, it is at these specific points that the transition from calcification to endochondral ossification can be analyzed [21]. In the present study, radiographic examination of the thyroid cartilage confirmed partial mineralization and showed calcification of two central points, posterior-superior and antero-inferior, during ossification of the structure (Figure 4).

This result demonstrated what occurs from the seventh stage of ossification de-growth onwards, concluding with the final stage. The ninth stage, as described above, corresponds to an age of approximately 57–68 years and, in the present case, the radiographic result confirms an anomaly in the development of cartilage belonging to a young soldier with a documented age of only 21 years.

Often, during a radiographic investigation, totally mineralized thyroid cartilage in very young subjects, such as the present case, can be mistaken for a foreign body both because of its rarity and because the calcified and non-oxidised posterior-superior and antero-inferior points would be radiotransparent under such an investigation [22]. Several variables could cause effective early mineralization of cartilage. Its elasticity prevents it from breaking, but at the same time, a trauma or contusion could produce small inflamed or hemorrhagic spots where inflammatory cells are recruited. This could lead to osteogenesis of the cartilage structure [23]. For this reason, more often than not, this ossification process is asymmetrical. Symmetrical or complete ossification could also depend on suspicion of metastatic calcification or on a generalized state of hyper-mineralization, generated by a metabolic disorder of the living subject.

In addition, disorders of mineral metabolism and other pathological degenerative processes, such as rheumatoid arthritis, have an important impact on total mineralization [24–26].

A decalcification test of the bone mineral component was performed to verify the ossification of the structure. Decalcification was achieved by placing the samples in 20% hydrochloric acid solutions for between three and five days [27]. The hydrochloric acid dissolved both calcium carbonate and calcium phosphate, the main components of hydroxyapatite, the inorganic part of the bone that gives rigidity and firmness. In the absence of these minerals, we would not have obtained any reaction on contact with the 20% acid.

During the anthropological analysis, several methods were applied to determine the age of the subject and among them was the method of the degree of fusion of the ossification nuclei of the entire skeletal system [12]. By averaging the degree of fusion of the ossification nuclei of the clavicle, humerus, and femur, the results determined a subject with a young age between 25–30 and also in this case, comparing the ossification nuclei of the thyroid cartilage under examination, an anomaly was reconfirmed as the result of its grade of fusion attests to an age between 57–68 years.

All the information obtained from the different methods applied in the study of this ossified thyroid cartilage supports the hypothesis of mineralization due to a physiopathological cause.

On the other hand, the skeletal system has many physiological variations that are not due to any cause that could lead us to a specific diagnosis [28]. Therefore, thyroid ossification, in this case, could also be considered a rare physiological variation.

## 4. Conclusions

Ossified thyroid cartilage in young subjects is a rare phenomenon that is often underestimated and ignored. The knowledge of this anatomical variation would be useful to interpret and determine different clinical conditions not considered in young patients.

In the present case, given the absence of clinical evidence about the cause of this abnormality, such as compressive trauma, hormonal dysfunction, or pathology that could lead to early mineralization, the hypothesis is that it may simply be a rare physiologic variation.

## 5. Impact Statement

The authors would like to thank the collaboration with the forensic physicians of the Department of Forensic Medicine of Bari (Apulia, Italy) and the Forensic Anthropology laboratory of the same department for the anthropological analysis performed on the collection of historical skeletal remains. In addition, the authors thank the anonymous reviewers for their comments.

**Author Contributions:** Conceptualization, F.I. and A.L.; methodology, F.I.; software, A.L.; validation, F.I., A.L.; formal analysis, A.L.; investigation, A.L.; resources, A.L.; data curation, F.I.; writing—original draft preparation, A.L.; writing—review and editing, F.I.; visualization, A.L.; supervision, F.I.; project administration, A.L.; funding acquisition, F.I. All authors have read and agreed to the published version of the manuscript.

**Funding:** This research received no external funding.

**Institutional Review Board Statement:** Not applicable.

**Informed Consent Statement:** Informed consent was obtained from all subjects involved in the study.

**Data Availability Statement:** Not applicable.

**Conflicts of Interest:** The authors declare no conflict of interest.

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
