# Peer review of "Fully Ossified Thyroid Cartilage Found among the Skeletal Remains of A 21-Year-Old Slavic Soldier: Interpretation of a Case"

_forensicsci, doi:10.3390/forensicsci1030019_

Round 1
Reviewer 1 Report
Dear Authors,
The case study presents an interesting case, but there are some points that need thorough
revision:
1. Line 33 – „ in the male than in the female sex “– use instead „in males than females “
2. Name Cenry – misspelled – correct is ÄŒerný (or Cerny) – please, carefully revise the whole text, including the figures and references, especially No. 7, 20. They are both identical, so use only one of them.
3. In fig. 2 add the description of c, d
4. Line 107 “The third investigation confirmed the results of the radiological
examination.” – be more specific, is it a decalcification test?
5. In the discussion section – were any particular pathological changes noted on the skeletal remains that could be associated with specific syndromes/ diseases/ trauma and etc.? Please explain any possible pathological changes.
6. In the conclusion, please provide a specific conclusion in objective terms e. g. how the study alone contributes to existing knowledge in the forensic field?
Author Response
Hello,
I wanted to thank the reviewers for their time spent on my manuscript.
REPLY COMMENTS:
Point 1. The sentence has been reworded following your requests.
Point 2. The name "Cenry" has been corrected and also the bibliographical notes as requested.
Point 3. Image descriptions (c,d) have been added.
Point 4. The sentence has been reworded to refer to the decalcification test which basically reconfirms the complete mineralisation of the cartilage analysed in the second test, the radiological test.
Point 5. In the paragraph of the Discussion, various hypotheses of pathology that could cause the early mineralisation are analysed but in the end a physiological variation is hypothesised given the absence of pathology that could have caused the ossification at a young age.
Point 6. The conclusion emphasises that cases like this are rare and that it would be necessary to study this type of study in depth to understand the origin of the anomaly described.
Reviewer 2 Report
The article presents an interesting case report of a complete and rather uncommon mineralization of the thyroid cartilage in a 21-years old subject. The authors perform an anthropological examination of the skeleton with a special emphasis on a multiapproach analysis of the ossified thyroid cartilage. The paper represents an important contribution to the subject of possible anatomical variations that can be useful at the moment of age estimation from skeletal material.
There are, however, minor aspects that raise my reservations.
On the page 3 (line 93) the authors use the word “implemented” to describe the use of additional age estimation methods (“degree of dental wear by dental analysis and the study of the morphology of the pubic symphysis”). Is it possible that they meant “completed”?
Initially, the authors state that the case report refers to the remains of a 21-years old soldier. Later on they inform that they will perform an anthropological analysis in order to “reconstruct the biological profile” (line 82) and this would include: “sex, age, stature and time of death” (line 83-84). Considering the fact that, in the past, it was rather uncommon to encounter a woman soldier and given the fact that the age of the individual was known a-priori, as stated by the authors, the “reconstruction” of the biological profile seems rather irrelevant. Especially, as in the results (case report section) and discussion part, the authors make no reference to the two remaining aspects they initially aimed at studying - stature and time of death. I would recommend a reformulation of this part, and more importantly, to include information about the results of stature and time of death in the text, if these aspects were indeed analyzed. Otherwise, I would recommend to remove them from the list of features to be analyzed. Maybe the term “reconstruction of biological profile” could be substituted by, for example, “analysis of biological profile” or “examination of biological profile” or “confirmation of sex and age at death”?
In the line 106 the authors state that the radiographic analysis “confirmed the total degree of ossification”, yet in the discussion part (line 142) they inform that the analysis “confirmed partial mineralization” of the cartilage. This is somehow confusing and I would recommend to unify this aspect.
In the discussion part, I miss addressing the results of the ossification of other skeletal elements - epiphyseal fusion - in relation to the hypothesis of hyper-mineralization, as such condition would probably affect the whole body. I believe this could be an interesting aspect to include in the paper’s discussion.
In my opinion, the discussion in general is somehow shallow and in a big part focuses on the results rather than their discussion. I believe the discussion could be improved.
Author Response
Hi,
I wanted to thank you for revising the manuscript and for taking the time to write it.
REPLY COMMENTS:
Point 1. Yes on Page 3 you wanted to state "Completed".
Point 2. Yes, a sex analysis was done on the skeletal remains because apparently there were also women among the skeletal remains of the soldiers analysed, most likely nurses. So for that reason an anthropological analysis of the biological profile was done for safety. With regard to the methods used in the analysis, the paragraph has been amended to add the method of stature and the determination of the time of death has been eliminated as it indicates 1946 as the date.
Point 3. The sentence has been changed to partial not total. It must have been a transcription error.
Points 4. and 5. The discussion has been modified and enriched following the requested indications.
I hope I have met the review criteria and I am available for any further information.
Thank you for your availability